# Regulation of Oncogenic Targets by *miR-99a-3p* (Passenger Strand of *miR-99a-*Duplex) in Head and Neck Squamous Cell Carcinoma

**DOI:** 10.3390/cells8121535

**Published:** 2019-11-28

**Authors:** Reona Okada, Keiichi Koshizuka, Yasutaka Yamada, Shogo Moriya, Naoko Kikkawa, Takashi Kinoshita, Toyoyuki Hanazawa, Naohiko Seki

**Affiliations:** 1Department of Functional Genomics, Chiba University Graduate School of Medicine, Chiba 260-8670, Japan; reonaokada@chiba-u.jp (R.O.); kkoshizuka@chiba-u.jp (K.K.); yasutaka1205@olive.plala.or.jp (Y.Y.); naoko-k@hospital.chiba-u.jp (N.K.); 2Department of Otorhinolaryngology/Head and Neck Surgery, Chiba University Graduate School of Medicine, Chiba 260-8670, Japan; t.kinoshita903@gmail.com (T.K.); thanazawa@faculty.chiba-u.jp (T.H.); 3Department of Biochemistry and Genetics, Chiba University Graduate School of Medicine, Chiba 260-8670, Japan; moriya.shogo@chiba-u.jp

**Keywords:** head and neck squamous cell carcinoma, microRNA, *miR-99a-3p*, passenger strand, antitumor, *STAMBP*

## Abstract

To identify novel oncogenic targets in head and neck squamous cell carcinoma (HNSCC), we have analyzed antitumor microRNAs (miRNAs) and their controlled molecular networks in HNSCC cells. Based on our miRNA signature in HNSCC, both strands of the *miR-99a*-duplex (*miR-99a-5p*: the guide strand, and *miR-99a-3p*: the passenger strand) are downregulated in cancer tissues. Moreover, low expression of *miR-99a-5p* and *miR-99a-3p* significantly predicts poor prognosis in HNSCC, and these miRNAs regulate cancer cell migration and invasion. We previously showed that passenger strands of miRNAs have antitumor functions. Here, we screened *miR-99a-3p*-controlled oncogenes involved in HNSCC pathogenesis. Thirty-two genes were identified as *miR-99a-3p*-regulated genes, and 10 genes (*STAMBP*, *TIMP4*, *TMEM14C*, *CANX*, *SUV420H1*, *HSP90B1*, *PDIA3*, *MTHFD2*, *BCAT1*, and *SLC22A15*) significantly predicted 5-year overall survival. Notably, among these genes, *STAMBP*, *TIMP4*, *TMEM14C*, *CANX*, and *SUV420H1* were independent prognostic markers of HNSCC by multivariate analyses. We further investigated the oncogenic function of *STAMBP* in HNSCC cells using knockdown assays. Our data demonstrated that the aggressiveness of phenotypes in HNSCC cells was attenuated by si*STAMBP* transfection. Moreover, aberrant STAMBP expression was detected in HNSCC clinical specimens by immunohistochemistry. This strategy may contribute to the clarification of the molecular pathogenesis of this disease.

## 1. Introduction

Head and neck squamous cell carcinoma (HNSCC) is the sixth most common cancer, with approximately 650,000 new cases diagnosed annually and 400,000 HNSCC-related deaths worldwide each year [1]. Tobacco and alcohol drinking habits are the major risk factors of HNSCC carcinogenesis [2]. In the past decade, our understanding of the role of human papillomavirus in the development of oropharyngeal squamous cell carcinoma has significantly changed the treatment strategy of this disease [3,4]. HNSCC is typically diagnosed when already at an advanced stage. Despite advancements in surgery, radiation therapy, and chemotherapy, patients with advanced HNSCC have a poor prognosis [1,4] owing to recurrence, metastasis, and treatment resistance [5]. The median overall survival time for patients with recurrence and metastasis is 10–13 months in the setting of first-line chemotherapy and 6 months in the second-line setting [6]. Recently, epidermal growth factor receptor inhibitors and immune checkpoint inhibitors have emerged as therapeutic approaches in HNSCC treatment [7,8]. However, these treatments do not yield satisfactory results.

MicroRNAs (miRNAs) exist widely in eukaryotes, and more than 2500 types of mature miRNAs have been discovered in humans [9,10]. miRNAs are transcribed from the human genome and then processed into mature miRNAs of approximately 18–22 bases [9,10]. miRNAs are classified as noncoding RNAs and function to suppress the translation of mRNAs by binding to the complementary sequence in the 3′-untranslated region (3′-UTR) of the targeted mRNA [9,10]. Notably, one miRNA targets multiple mRNAs, and there are multiple miRNA binding sites in the UTRs of one mRNA [9,10]. Therefore, changes in the expression of miRNAs are involved in various diseases, including human cancers, suggesting that miRNAs play important roles in disease development [11,12,13,14,15].

We have been studying antitumor miRNAs and their oncogenic networks in HNSCC cells based on HNSCC miRNA signatures [16,17,18,19,20]. Our previous studies have shown that the antitumor *miR-29* family directly controls laminin-332 and integrins (*ITGA6*, *ITGB4*, and *ITGB1*), and *miR-199* family targets *ITGA3* in HNSCC cells [21,22,23]. Moreover, the antitumor *miR-26* family, *miR-29* family, and *miR-218* inhibit cancer cell migration and invasion in HNSCC cells, and these miRNAs coordinately regulate lysyl oxidase like 2 [24]. These antitumor miRNAs (the *miR-26* family, the *miR-29* family, *miR-218*, and the *miR-199* family) target proteins involved in the epithelial-mesenchymal transition, indicating their pivotal roles in metastasis in cancer cells.

In this study, we focused on *miR-99a-5p* (the guide strand of the *miR-99a*-duplex) and *miR-99a-3p* (the passenger strand) based on our HNSCC miRNA signature determined by RNA sequencing [20]. Previous studies have shown that downregulation of *miR-99a-5p* occurs in various cancers and that the expression of this miRNA attenuates malignant phenotypes in cancer cells, suggesting that *miR-99a-5p* acts as an antitumor miRNA [25,26]. However, few reports have described the roles of the passenger strand *miR-99a-3p* in HNSCC, and oncogenic networks controlled by *miR-99a-3p* are still unknown. In the general concept of miRNA biogenesis, passenger strands of miRNAs are degraded in the cytosol and have no function [9,10]. However, our previous studies showed that some passenger strands of miRNAs, e.g., *miR-145-3p, miR-150-3p,* and *miR-199a/b-3p* were downregulated in the signature and acted as antitumor miRNAs in malignant cells. Importantly, several targets regulated by these passenger strands of miRNAs acted as oncogenes, and their aberrant expressions were closely associated with the poor prognosis of the patients [23,27,28,29,30]. Therefore, the analysis of passenger strands of miRNAs is useful for understanding the molecular pathogenesis of HNSCC.

Our functional assays indicated that ectopic expression of both strands of the *miR-99a*-duplex significantly attenuated malignant phenotypes in HNSCC cells. We further analyzed *miR-99a-3p*-regulated oncogenic genes involved in HNSCC molecular pathogenesis. In total, 32 genes were identified as *miR-99a-3p*-controlled genes, and 10 genes (*STAMBP*, *TIMP4*, *TMEM14C*, *CANX*, *SUV420H1*, *HSP90B1*, *PDIA3*, *MTHFD2*, *BCAT1*, and *SLC22A15*) significantly predicted 5-year overall survival in patients with HNSCC. Moreover, our findings revealed that aberrant expression of *STAMBP* enhanced cancer cell aggressiveness in HNSCC.

## 2. Materials and Methods

### 2.1. Clinical Human HNSCC Specimens and HNSCC Cell Lines

Twenty-two clinical specimens were obtained from patients with HNSCC following surgical tumor resection at Chiba University Hospital (2008–2013, Chiba, Japan). The patients’ clinical characteristics are shown in Table 1. Written informed consent was obtained from all patients before the use of their specimens. This study was approved by the Bioethics Committee of Chiba University (approval number: 811(690)). Normal tissue was collected from the most distant cancerous part of the same specimen. A total of 22 pairs of HNSCC tissues and adjacent normal (noncancerous) tissues were obtained in this study.

Two HNSCC cell lines, FaDu (American Type Culture Collection, Manassas, VA, UAS) and SAS cells (RIKEN Cell Bank, Tsukuba, Ibaraki, Japan), were used in this study.

### 2.2. RNA Extraction and Quantitative Real-Time Reverse Transcription Polymerase Chain Reaction (qRT-PCR)

RNA was extracted from clinical specimens and cell lines as previously described [20,23,30,31,32]. miRNA expression levels were evaluated using qRT-PCR as described previously [20,23,30,31,32]. The TaqMan probes and primers used in this study are listed in Appendix A.

### 2.3. Transfection of miRNAs, siRNAs, and Plasmid Vectors into HNSCC Cells

The procedures for transfection of miRNAs, siRNAs, and plasmid vectors into HNSCC cells were described previously [20,23,30,31,32]. The reagents used in this study are listed in Appendix A.

### 2.4. Functional Assays in HNSCC Cells (Cell Proliferation, Migration and Invasion Assays)

The procedures for functional assays in cancer cells (proliferation, migration, and invasion) are described in our previous studies [20,23,30,31,32]. Cells were transfected with 10nM miRNAs or siRNAs. Cell proliferation was evaluated with XTT assays. Migration assays were performed with uncoated transwell polycarbonate membrane filters, invasion assays with modified Boyden chambers.

### 2.5. Measurement of miR-99a-3p Incorporated into the RISC

Immunoprecipitation using anti-Ago2 antibodies was performed to determine whether *miR-99a-3p* was incorporated into the RISC. FaDu and SAS were transfected with 10nM miRNAs for 48 h and the collected cells went through immunoprecipitation using human anti-Ago2 antibodies (microRNA Isolation Kit, Human Ago2; Wako, Osaka, Japan) according to the manufacture’s protocol. Obtained miRNAs proceeded to qRT-PCR. For normalization of the results, *miR-26a* was measured, whose expression was not affected by *miR-99a-5p/3p* transfection. The procedure for immunoprecipitation was described in previous studies [23,30,31,32]. The reagents used in this study are listed in Appendix A.

### 2.6. Identification of miR-99a-3p and miR-99a-5p Targets in HNSCC Cells

The strategy for identification of miRNA targets in this study is summarized in Appendix A. Two expression profiles (i.e., *miR-99a-5p*-transfected FaDu cells [GEO accession number: GSE123318], *miR-99a-3p*-transfected FaDu cells [accession number: GSE123318]) were used in this screening. The TargetScanHuman database (http://www.targetscan.org/vert_72/) was used to predict miRNA binding sites.

### 2.7. Plasmid Construction and Dual-Luciferase Reporter Assays

Plasmid vectors, including vectors containing the wild-type sequences of *miR-99a-3p* binding sites in the 3′-UTR of *STAMBP* or the deletion sequences of *miR-99a-3p* binding sites in the 3′-UTR of *STAMBP*, were prepared. The inserted sequences are shown in Appendix A. The procedures for transfection and dual luciferase reporter assays were described in our previous studies [20,23,30,31,32]. The reagents used in this study are listed in Appendix A.

### 2.8. Clinical Data Analyses of miRNAs and Target Genes in HNSCC Specimens

TCGA (https://tcga-data.nci.nih.gov/tcga/) was applied to investigate the clinical significance of miRNAs and their target genes. Gene expression and clinical data were obtained from cBioPortal (http://www.cbioportal.org/) and OncoLnc (http://www.oncolnc.org/) (data downloaded on 1 August 2019).

### 2.9. Western Blotting and Immunohistochemistry

The procedures for Western blotting and immunohistochemistry were described in our previous studies [20,23,30,31,32]. The antibodies used in this study are listed in Appendix A.

### 2.10. Statistical Analyses

Mann–Whitney U tests were applied for comparisons between two groups. For multiple groups, one-way analysis of variance and Tukey tests for post-hoc analysis were applied. These analyses were performed with JMP Pro 14 (SAS Institute Inc., Cary, NC, USA).

## 3. Results

### 3.1. Downregulation and Clinical Significance of miR-99a-5p and miR-99a-3p in HNSCC Clinical Specimens

The clinical features of HNSCC specimens are listed in Table 1. Expression levels of *miR-99a-5p* and *miR-99a-3p* were significantly low in cancer tissues compared with those in normal tissues from the same patients (*p* < 0.0001 and *p* < 0.0001, respectively; Figure 1A and Appendix A). The expression levels of these miRNAs in two HNSCC cell lines (FaDu and SAS cells) were also very low compared with those in normal tissues (Figure 1A and Appendix A). A positive correlation was detected between *miR-99a-5p* and *miR-99a-3p* expression levels by Spearman’s rank analysis (R = 0.716, *p* < 0.0001; Figure 1B).

Cohort analysis using data from The Cancer Genome Atlas (TCGA) database revealed that low expression of *miR-99a-5p* and *miR-99a-3p* was associated with poorer survival in patients with HNSCC (*p* = 0.0008 and *p* = 0.0012, respectively; Figure 1C). We confirmed positive correlation of these miRNAs expression by using TCGA database sets (Appendix A).

### 3.2. Ectopic Expression of miR-99a-5p and miR-99a-3p on Cell Proliferation, Migration and Invasion in HNSCC Cells

To investigate the anti-tumor functions of *miR-99a-5p* and *miR-99a-3p* in HNSCC cells, we assessed changes in cell proliferation, migration, and invasion after transfection of these miRNAs into FaDu and SAS cells. Notably, ectopic expression of *miR-99a-5p* significantly decreased cell proliferation (Figure 2A). However, cell proliferation was not affected by *miR-99a-3p* transfection. Additionally, the migration and invasion of FaDu and SAS cells were significantly suppressed by *miR-99a-5p* and *miR-99a-3p* transfection (Figure 2B,C). Photomicrographs are presented in Appendix A.

### 3.3. Incorporation of miR-99a-5p and miR-99a-3p into the RNA-Induced Silencing Complex (RISC) in HNSCC Cells

Ago2 is an essential component of the RISC. Therefore, to verify that *miR-99a-5p* and *miR-99a-3p* had functions in HNSCC cells, immunoprecipitation assays were carried out using anti-Ago2 antibodies. After transfection of both miRNAs to SAS cells, the amounts of *miR-99a-5p* and *miR-99a-3p* were significantly increased relative to that in control (untransfected) cells (Appendix A). These data showed that *miR-99a-5p* (the guide strand) and *miR-99a-3p* (the passenger strand) were both incorporated into the RISC in HNSCC cells.

### 3.4. Screening of Molecular Targets Regulated by miR-99a-5p and miR-99a-3p in HNSCC Cells

To identify the genes controlled by *miR-99a-5p* and *miR-99a-3p* in HNSCC cells, we used gene expression data obtained by RNA microarray analysis of *miR-99a-5p*- or *miR-99a-3p*-transfected FaDu cells and data from the TargetScanHuman database (release 7.2), which provided annotated putative targets for each miRNA. Our strategy searching for *miR-99a-5p* and *miR-99a-3p* target genes is shown in Appendix A.

Using this strategy, only genes from *miR-99a-3p*-transfected FaDu cells endured the selection process. For *miR-99a-3p*, 114 genes were identified as putative target genes in HNSCC cells (Table 2). Eighteen genes were identified as *miR-99a-5p*-controlled genes, none of which showed correlations with prognosis in TCGA database (Appendix A).

### 3.5. Clinical Significance of miR-99a-3p Targets in HNSCC Pathogenesis

By using TCGA database, we narrowed down the list of 114 genes according to correlations with 5-year overall survival rates. Among the genes, high expression of 10 genes (*STAMBP, TIMP4, TMEM14C, CANX, SUV420H1, HSP90B1, PDIA3, MTHFD2, BCAT1,* and *SLC22A15*) was associated with poor prognosis (5-year overall survival rate: *p* < 0.05) in patients with HNSCC (Table 2 and Figure 3). Furthermore, multivariate analysis elucidated that expression levels of five genes (*STAMBP, TIMP4, TMEM14C, SUV420H1,* and *CANX*) were independent prognostic factors for 5-year overall survival in these patients (Figure 4).

### 3.6. Direct Regulation of STAMBP by miR-99a-3p in HNSCC Cells

In cells transfected with *miR-99a-3p*, the levels of *STAMBP* mRNA and STAMBP protein were significantly lower than in mock- or miR-control-transfected cells (Figure 5A,B). The whole pictures of Western blotting are shown in Appendix A. In addition, we investigated whether the other four genes (*TIMP4, TMEM14C, SUV420H1,* and *CANX*) were controlled by *miR-99a-3p* in HNSCC cells at the RNA levels. Consistent with the estimation of TargetScanHuman database, the expression levels of the four genes were also downregulated by *miR-99a-3p* transfection in HNSCC cells (Appendix A).

Next, we performed dual-luciferase reporter assays to determine whether *STAMBP* was directly regulated by *miR-99a-3p*. We used vectors encoding the partial wild-type sequences of the 3′-UTR of *STAMBP*, including the predicted *miR-99a-3p* target site deletion vector lacking the *miR-99a-3p* binding site (Figure 5C and Appendix A). We found that luciferase activity was significantly decreased by cotransfection with *miR-99a-3p* and the vector carrying the wild-type 3′-UTR of *STAMBP*, whereas transfection with the deletion vector blocked the decrease in luminescence in FaDu and SAS cells (Figure 5D). These data demonstrated that *miR-99a-3p* directly bound to the 3′-UTR of *STAMBP*.

### 3.7. Effects of STAMBP Knockdown on Cell Proliferation, Migration, and Invasion in HNSCC Cells

To investigate the oncogenic functions of *STAMBP* in HNSCC cells, knockdown assays were conducted using small interfering RNAs (siRNAs). Both mRNA and protein expression levels were successfully suppressed by si*STAMBP*-1 and si*STAMBP*-2 transfection into FaDu and SAS cells (Figure 6A,B). The whole pictures of Western blotting are shown in Appendix A.

In functional assays, cell proliferation was not suppressed by si*STAMBP* transfection into FaDu cells. Besides, in SAS cells, cell proliferation was significantly suppressed by si*STAMBP* transfection (Figure 6C). Cell migration and invasive abilities were significantly blocked by knockdown of *STAMBP* (si*STAMBP*-1 and si*STAMBP*-2) in FaDu and SAS cells (Figure 6D,E). The photomicrographs are shown in Appendix A. Regarding the cell proliferation assay, the results differed between FaDu cells and SAS cells. To explain this phenomenon, a detailed analysis of genes involved in cell cycle and cell division for two cell lines will be necessary.

### 3.8. Overexpression of STAMBP in HNSCC Clinical Specimens

Expression of STAMBP protein was evaluated using HNSCC clinical specimens. Overexpression of STAMBP was detected in cancer lesions in HNSCC clinical specimens (Figure 7A–H). In contrast to cancer lesions, expression of STAMBP was extremely weak in normal mucosa (Figure 7J). Information on clinical specimens used for immunostaining is shown in Table 3.

To confirm our immunostaining results, we analyzed gene expression data of GEO database (accession number: GSE6631). Analysis of gene expression data showed that expression of *STAMBP* was significantly upregulated in HNSCC clinical specimens (Appendix A).

## 4. Discussion

Owing to the high rate of recurrence and metastasis in HNSCC, HNSCC is still a deadly cancer, with an average 50% overall 5-year survival rate [1,2,3,4,5,6]. In order to improve treatment outcomes in patients with HNSCC, it is essential to develop treatments for cases with recurrence and metastasis. Advanced genomic approaches are effective for elucidating the molecular pathogenesis of HNSCC, leading to the identification of molecular targets for treatment.

As part of the unique biological nature of miRNAs, a single miRNA can control (directly or indirectly) many RNA transcripts in each cell. Therefore, the aberrant expression of miRNA influences multiple pathways, including cell proliferation, migration, invasion, and apoptosis. Aberrantly expressed miRNAs disrupt RNA expression networks, resulting in cancer cell initiation, development, metastasis, and drug resistance [11,12,13,14,15]. Accordingly, we have sequentially identified antitumor miRNAs and their controlled molecular targets and pathways in HNSCC cells based on miRNA signatures [16,17,18,19,20]. We recently created an HNSCC miRNA expression signature by RNA sequencing [20]. Notably, our signatures revealed that some miRNA passenger strands, e.g., *miR-143-5p*, *miR-145-3p*, *miR-150-3p*, *miR-199a-3p*, and *miR-199b-3p* were downregulated in HNSCC tissues and that their expression status was closely involved in HNSCC molecular pathogenesis [20,23,27]. More recently, our group revealed that passenger strands of miRNAs exert antitumor roles by targeting several oncogenes in prostate cancer, renal cell carcinoma, esophageal squamous cell carcinoma, and lung cancer [28,29,30,31,32,33,34]. The participation of passenger strands of miRNAs in carcinogenesis is a new concept in cancer research.

In this study, we revealed that both strands of the *miR-99a*-duplex (*miR-99a-5p* and *miR-99a-3p*) acted as antitumor miRNAs in HNSCC cells. Moreover, we showed that 18 and 30 genes were putative targets of *miR-99a-5p* and *miR-99a-3p* regulation, respectively. Many studies have shown that *miR-99a-5p* acts as an antitumor miRNA in various cancers by targeting many oncogenes [35,36,37,38,39]. In contrast to *miR-99a-5p*, few papers have analyzed the functional significance of *miR-99a-3p* in cancer cells. Previously, downregulation of *miR-99a-3p* was detected in castration-resistant prostate cancer (CRPC), and ectopic expression of *miR-99a-3p* was found to attenuate cancer cell aggressive phenotypes [40]. Moreover, *miR-99a-3p* was shown to regulate non-SMC condensin I complex subunit G directly, and its overexpression was detected in CRPC clinical specimens, showing a significant association with shorter disease-free survival and advanced clinical stage [40]. In renal cell carcinoma cells, *miR-99a-3p* significantly inhibits cell proliferation and colony formation through regulating ribonucleotide reductase regulatory subunit-M2 [41]. More recently, lower expression of *miR-99a-3p* and its mediated molecular pathways were detected in HNSCC by in silico analysis, TCGA database, and Genotype-Tissue Expression sequencing databases [42]. These studies indicated that the downregulation of *miR-99a-3p* was closely involved in cancer pathogenesis.

In this study, we aimed to identify oncogenic targets regulated by *miR-99a-3p* in HNSCC cells. In total, 114 genes were identified as *miR-99a-3p*-controlled genes, and 10 of these genes (*STAMBP*, *TIMP4*, *TMEM14C*, *CANX*, *SUV420H1*, *HSP90B1*, *PDIA3*, *MTHFD2*, *BCAT1*, and *SLC22A15*) significantly predicted 5-year overall survival. Notably, among these genes, *STAMBP*, *TIMP4*, *TMEM14C*, *CANX*, and *SUV420H1* were independent prognostic markers of HNSCC, as demonstrated by multivariate analyses. Our preliminary analysis has shown that these genes were controlled by *miR-99a-3p* in HNSCC cells (Appendix A). Further detailed examinations are necessary in the future. A previous study showed that TIMP4 secretion was regulated by the expression of *LOX/SNAI2* axis and contributed to the malignant phenotype of cancers, e.g., thyroid cancer, colon cancer, and breast cancer [43]. Calnexin *CANX* is an integral protein of the endoplasmic reticulum and acts as a chaperon. In colorectal cancer, overexpression of *CANX* predicted poor prognosis of the patients, and its knockdown attenuated aggressive phenotypes of cancer cells [44]. Another study showed that serum levels of CANX were significantly higher in patients with lung cancer, and its expression was a useful sero-diagnostic marker of the patients [45]. Overexpression of *SUV420H1* (acts as lysine methyl transferase) enhanced oncogenic *ERK* signaling through *ERK* phosphorylation and transcription [46]. These genes may be candidate prognostic markers and therapeutic targets in HNSCC. Functional analysis of these genes will reveal new molecular pathologies for HNSCC.

Among these targets, we further investigated the oncogenic roles of *STAMBP* in HNSCC cells. STAM-binding protein (STAMBP) is a deubiquitinating enzyme that interacts with the SH3 domain of STAM. This protein plays key roles in cell surface receptor-mediated endocytosis and sorting and in cytokine-mediated signaling for MYC induction and cell cycle progression [47,48,49,50,51]. Whole-exome sequencing revealed that the microcephaly-capillary malformation syndrome was related to recessive mutations in *STAMBP* [52]. In cancer research, almost no functional analysis of *STAMBP* has been conducted. A recent study showed that *STAMBP* expression contributes to melanoma cell migration and invasion through the stabilization of SLUG expression [53]. This result was consistent with our current HNSCC data. In this study, overexpression of STAMBP was detected in HNSCC clinical specimens; knockdown assays using siRNAs demonstrated that migration and invasion were significantly reduced in HNSCC cells. Thus, overexpression of *STAMBP* may promote cancer cell metastasis. Further studies are needed to analyze the molecular networks controlled by *STAMBP* in various cancers.

## 5. Conclusions

Based on the miRNA expression signature of HNSCC, we revealed that *miR-99a-3p* (the passenger strand) acted as an antitumor miRNA in HNSCC cells. In total, 10 genes (*STAMBP, TIMP4, TMEM14C, CANX, SUV420H1, HSP90B1, PDIA3, MTHFD2, BCAT1,* and *SLC22A15*) were regulated by *miR-99a-3p* in HNSCC cells and were closely involved in HNSCC molecular pathogenesis. *STAMBP* expression was directly controlled by *miR-99a-3p*, and its overexpression enhanced cancer cell migration and invasion. Our strategy, i.e., identification of antitumor miRNAs and their targets, may be an attractive tool to reveal novel prognostic and therapeutic targets in HNSCC.

## Figures and Tables

**Figure 1 cells-08-01535-f001:**
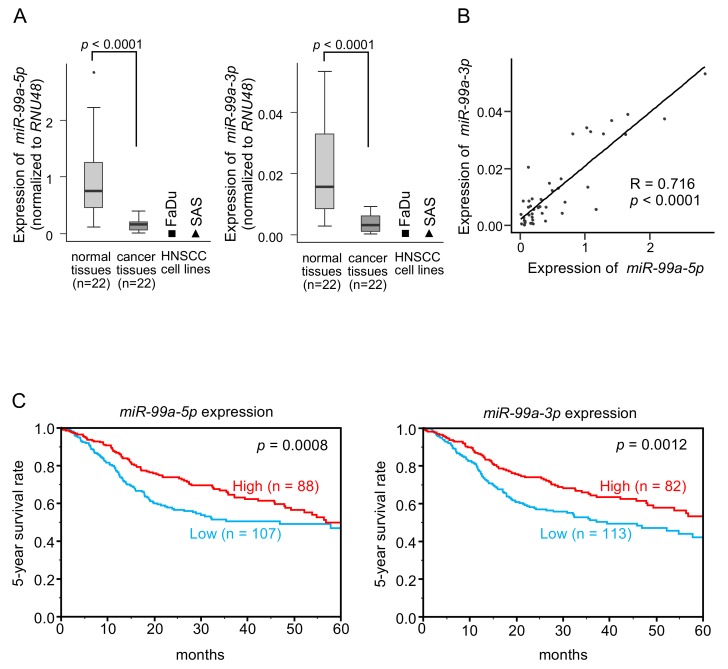
Expression and clinical significance of *miR-99a-5p* and *miR-99a-3p* in HNSCC clinical specimens. (**A**) Expression of *miR-99a-5p* and *miR-99a-3p* was significantly reduced in HNSCC clinical specimens and cell lines (FaDu and SAS cells). Data were normalized to the expression of RNU48. (**B**) Spearman’s rank tests showed positive correlations between expression levels of *miR-99a-5p* and *miR-99a-3p* in clinical specimens. (**C**) Kaplan-Meier survival curve analyses of patients with HNSCC using data from The Cancer Genome Atlas (TCGA) database. Patients were divided into two groups according to miRNA expression, high group and low group (according to median expression). The red line shows the high expression group, and the blue line shows the low expression group.

**Figure 2 cells-08-01535-f002:**
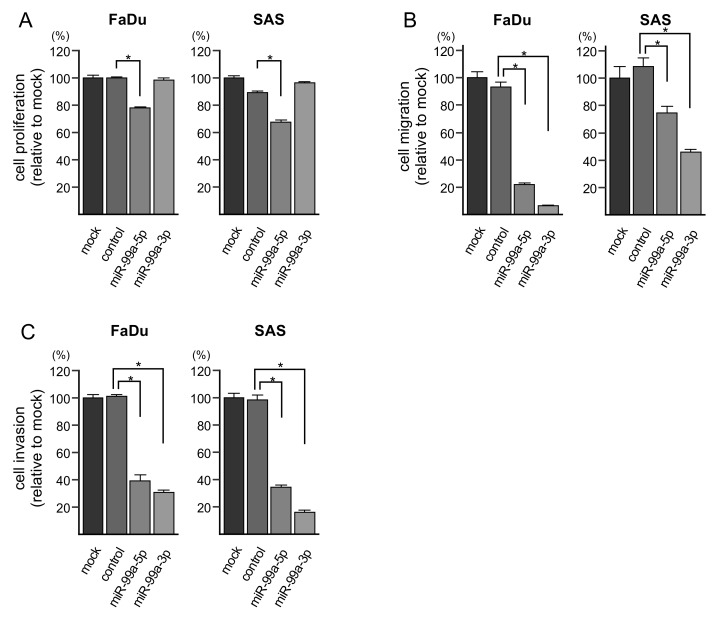
Functional assays of cell proliferation, migration, and invasion following ectopic expression of *miR-99a-5p* and *miR-99a-3p* in HNSCC cell lines (FaDu and SAS cells). (**A**) Cell proliferation was assessed using XTT assays. Data were collected 72 h after miRNA transfection (* *p* < 0.0001). (**B**) Cell migration was assessed with membrane culture system. Data were collected 48 h after seeding the cells into the chambers (* *p* < 0.0001). (**C**) Cell invasion was determined 48 h after seeding miRNA-transfected cells into chambers using Matrigel invasion assays (* *p* < 0.0001).

**Figure 3 cells-08-01535-f003:**
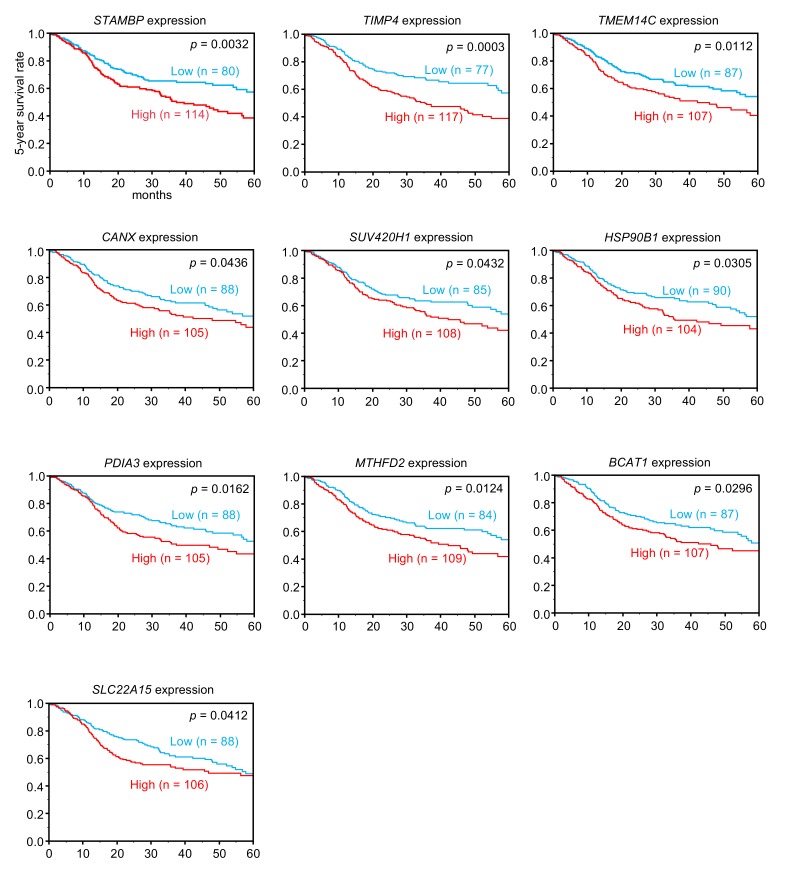
Clinical significance of *miR-99a-3p* target genes in TCGA database. Among putative targets of *miR-99a-3p* in HNSCC cells, high expression of 10 genes (*STAMBP, TIMP4, TMEM14C, CANX, SUV420H1, HSP90B1, PDIA3, MTHFD2, BCAT1,* and *SLC22A15*) was significantly associated with poor prognosis in patients with HNSCC. Kaplan-Meier curves of 5-year overall survival for each gene are shown.

**Figure 4 cells-08-01535-f004:**
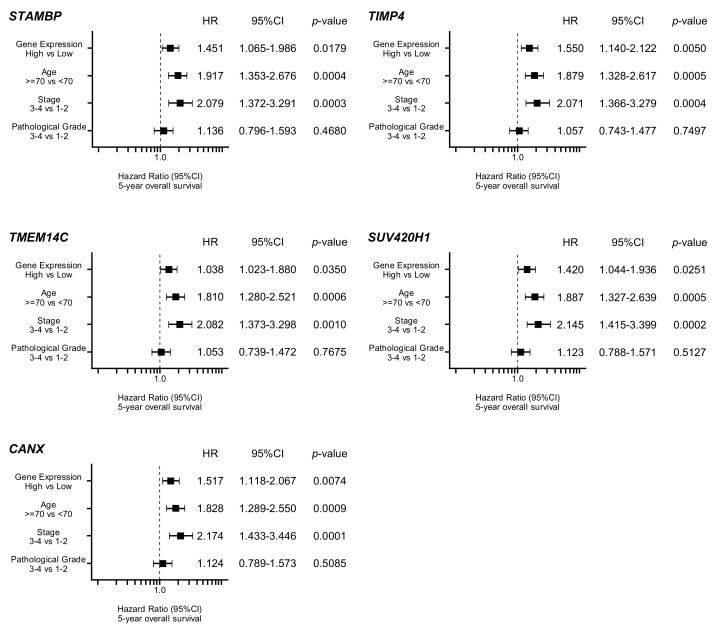
Forest plot of multivariate analysis of five genes (*STAMBP, TIMP4, TMEM14C, SUV420H1,* and *CANX*), which were independent prognostic factors for overall survival after adjustment for patient age, disease, stage, and pathological grade.

**Figure 5 cells-08-01535-f005:**
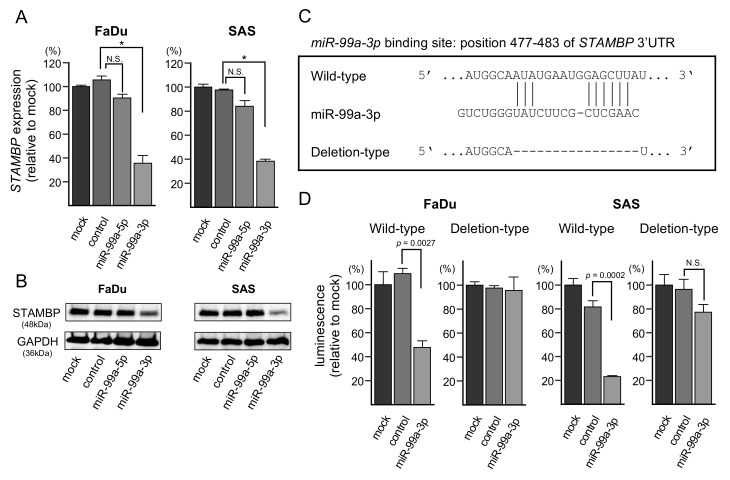
Expression of *STAMBP*/STAMBP was directly regulated by *miR-99a-3p* in HNSCC cells. (**A**) Expression of *STAMBP* mRNA was significantly reduced by *miR-99a-3p* transfection into FaDu and SAS cells (72 h after transfection; * *p* < 0.0001, N.S.: Not significant). Expression of *GAPDH* was used as an internal control. (**B**) Expression of STAMBP protein was reduced by *miR-99a-3p* transfection into HNSCC cells (72 h after transfection). Expression of GAPDH was used as an internal control. (**C**) TargetScanHuman database analyses predicted one putative *miR-99a-3p* binding site in the 3′-UTR of *STAMBP*. (**D**) Dual luciferase reporter assays showed that luminescence activities were reduced by cotransfection with wild-type (*miR-99a-3p* binding site) vectors and *miR-99a-3p* in FaDu and SAS cells. Normalized data were calculated as Renilla/firefly luciferase activity ratios (N.S.: Not significant).

**Figure 6 cells-08-01535-f006:**
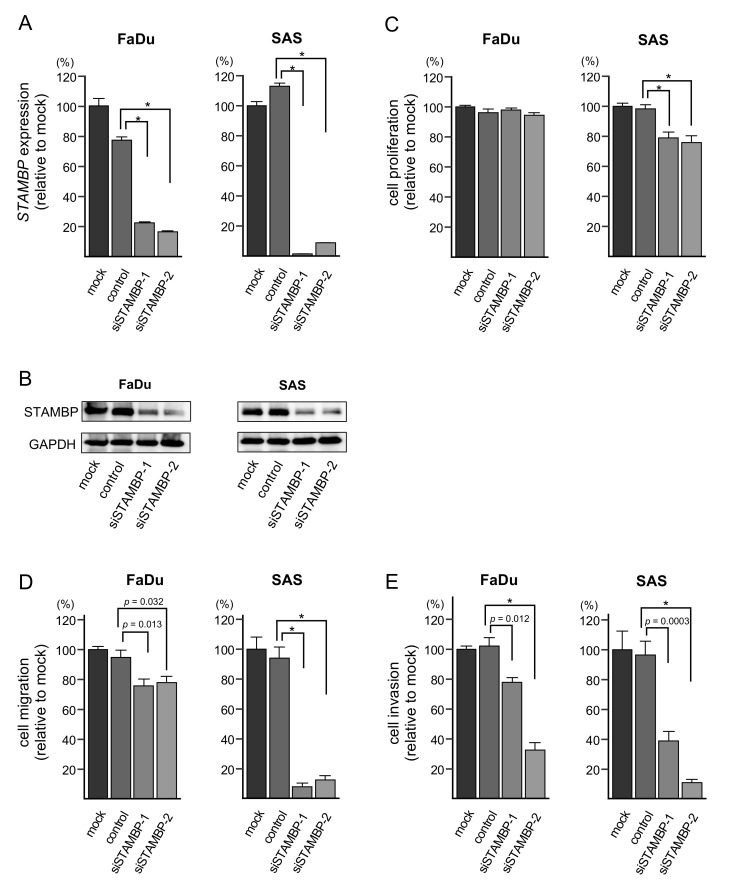
Effects of *STAMBP* knockdown on cell proliferation, migration, and invasion in HNSCC cells. (**A**) Expression of *STAMBP* mRNA was significantly reduced by siRNA transfection into HNSCC cells (* *p* < 0.0001). Expression of *GAPDH* was used as an internal control. (**B**) Expression of STAMBP protein was markedly reduced by siRNA transfection into HNSCC cells. Expression of GAPDH was used as an internal control. (**C**) Cell proliferation was assessed using XTT assays. Data were collected 72 h after miRNA transfection (* *p* < 0.0001). (**D**) Cell migration was assessed with a membrane culture system. Data were collected 48 h after seeding the cells into the chambers (* *p* < 0.0001). (**E**) Cell invasion was determined 48 h after seeding miRNA-transfected cells into chambers using Matrigel invasion assays (* *p* < 0.0001).

**Figure 7 cells-08-01535-f007:**
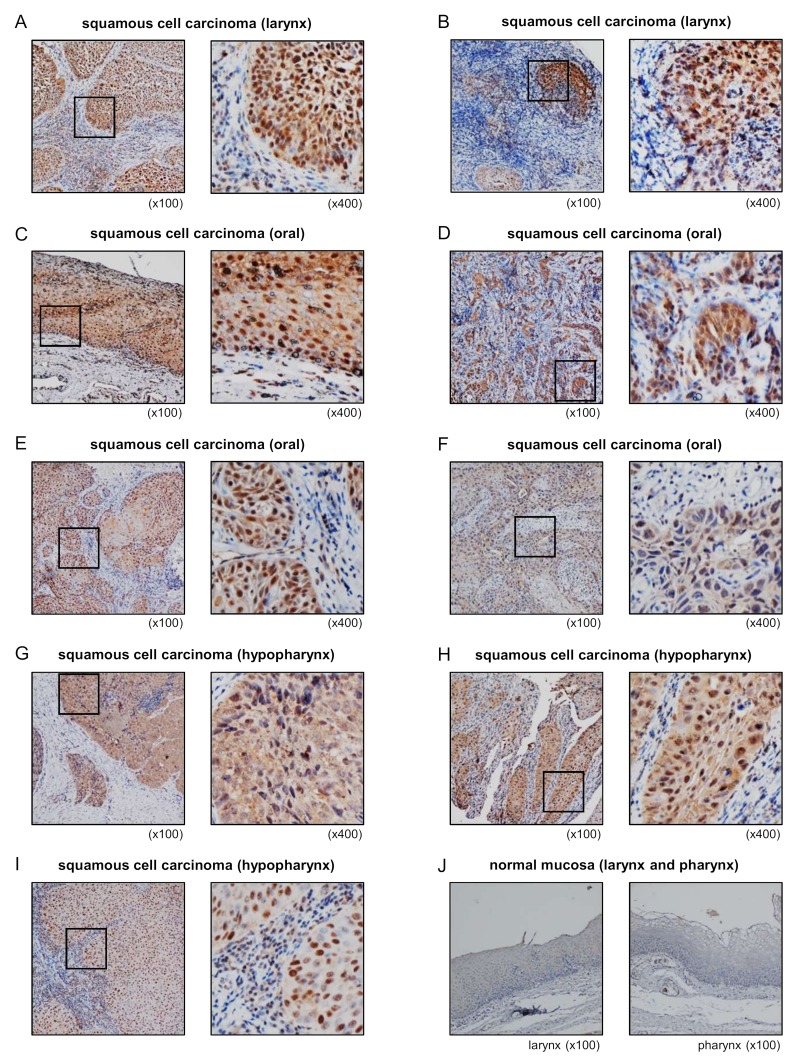
Overexpression of STAMBP in HNSCC clinical specimens. (**A**–**I**) Expression of STAMBP was investigated by immunohistochemical staining of HNSCC clinical specimens. Overexpression of STAMBP was detected in the nuclei and/or cytoplasm of cancer cells. (**J**) Extremely weak expression of STAMBP in normal mucosa of larynx and pharynx.

**Table 1 cells-08-01535-t001:** Clinical features of 22 HNSCC patients.

No.	Age	Sex	Location	T	N	M	Stage	Differentiation
1	66	M	hypopharynx	2	2c	0	IVa	moderate
2	66	M	hypopharynx	4a	2c	0	IVa	well
3	66	M	hypopharynx	4b	2c	0	IVb	moderate
4	76	M	hypopharynx	4a	1	0	IVa	well
5	74	M	hypopharynx	4a	2c	0	IVa	poor
6	45	M	hypopharynx	4a	2c	0	IVa	moderate
7	75	M	hypopharynx	4a	2c	0	IVa	well
8	58	M	hypopharynx	4a	0	0	IVa	well
9	69	M	larynx	3	0	0	III	well
10	70	M	larynx	4a	1	0	IVa	well-moderate
11	84	M	larynx	4a	0	0	IVa	moderate
12	50	M	larynx	4a	2b	0	IVa	moderate
13	82	M	larynx	4a	0	0	IVa	moderate
14	85	M	larynx	3	2b	0	IVa	moderate
15	66	M	tongue	2	0	0	II	moderate
16	73	M	tongue	3	1	0	III	poor
17	74	M	tongue	1	0	0	I	well
18	72	M	tongue	4a	2b	0	IVa	moderate
19	83	M	oral floor	2	1	0	III	well
20	68	F	oral floor	4a	1	0	IVa	well
21	77	M	oral floor	2	2b	0	IVa	moderate
22	69	M	oropharynx	1	0	0	I	well

T: Primary tumor stage, N: Regional lymph nodes stage, M: Distant metastasis stage. All according to the UICC (The Union for International Cancer Control) classification.

**Table 2 cells-08-01535-t002:** Candidate target genes regulated by *miR-99a-3p.*

Entrez Gene ID	Gene Symbol	Gene Name	Total Sites	FaDu *miR-99a-3p* Transfectant FC (log2)	TCGA OncoLnc 5-Year OS *p*-Value
10617	*STAMBP*	STAM binding protein	1	−1.0548	0.0032
7079	*TIMP4*	TIMP metallopeptidase inhibitor 4	1	−2.2976	0.0003
51522	*TMEM14C*	transmembrane protein 14C	1	−3.0740	0.0112
51111	*SUV420H1 (KMT5B)*	suppressor of variegation 4-20 homolog 1	1	−1.0238	0.0432
821	*CANX*	calnexin	1	−2.3590	0.0436
10797	*MTHFD2*	methylenetetrahydrofolate dehydrogenase (NADP+ dependent) 2,methenyltetrahydrofolate cyclohydrolase	1	−1.6298	0.0124
2923	*PDIA3*	protein disulfide isomerase family A, member 3	1	−1.3332	0.0162
586	*BCAT1*	branched chain amino-acid transaminase 1, cytosolic	3	−2.0236	0.0296
7184	*HSP90B1*	heat shock protein 90kDa beta (Grp94), member 1	1	−2.3549	0.0305
55356	*SLC22A15*	solute carrier family 22, member 15	1	−1.9007	0.0412
23786	*BCL2L13*	BCL2-like 13 (apoptosis facilitator)	1	−1.2622	0.0604
29967	*LRP12*	low density lipoprotein receptor-related protein 12	1	−1.0258	0.0897
112752	*IFT43*	intraflagellar transport 43	1	−1.4595	0.0922
23516	*SLC39A14*	solute carrier family 39 (zinc transporter), member 14	1	−1.6435	0.1124
55255	*WDR41*	WD repeat domain 41	1	−1.2410	0.1145
56886	*UGGT1*	UDP-glucose glycoprotein glucosyltransferase 1	1	−1.0232	0.1228
6137	*RPL13*	ribosomal protein L13	1	−1.6427	0.1408
114971	*PTPMT1*	protein tyrosine phosphatase, mitochondrial 1	1	−1.2094	0.1545
27	*ABL2*	ABL proto-oncogene 2, non-receptor tyrosine kinase	1	−1.8956	0.1549
114818	*KLHL29*	kelch-like family member 29	1	−1.4701	0.1551
2512	*FTL*	ferritin, light polypeptide	1	−1.2722	0.1809
84803	*AGPAT9 (GPAT3)*	1-acylglycerol-3-phosphate O-acyltransferase 9	1	−1.8971	0.2008
23271	*CAMSAP2*	calmodulin regulated spectrin-associated protein family, member 2	1	−1.0735	0.3277
122953	*JDP2*	Jun dimerization protein 2	1	−2.2594	0.3311
219902	*TMEM136*	transmembrane protein 136	1	−1.4956	0.3455
440026	*TMEM41B*	transmembrane protein 41B	1	−2.3021	0.3843
54629	*FAM63B*	family with sequence similarity 63, member B	1	−1.2972	0.3940
182	*JAG1*	jagged 1	1	−1.0082	0.3971
2121	*EVC*	Ellis van Creveld syndrome	1	−1.7233	0.4006
490	*ATP2B1*	ATPase, Ca++ transporting, plasma membrane 1	1	−2.0436	0.5557
9208	*LRRFIP1*	leucine rich repeat (in FLII) interacting protein 1	1	−1.1882	0.6483
50848	*F11R*	F11 receptor	1	−1.0283	0.7312
79152	*FA2H*	fatty acid 2-hydroxylase	1	−1.7458	0.0877
23049	*SMG1*	SMG1 phosphatidylinositol 3-kinase-related kinase	1	−1.1426	0.1267
5337	*PLD1*	phospholipase D1, phosphatidylcholine-specific	1	−1.0968	0.1538
5935	*RBM3*	RNA binding motif (RNP1, RRM) protein 3	1	−1.3755	0.1554
135398	*C6orf141*	chromosome 6 open reading frame 141	1	−1.4022	0.1594
5251	*PHEX*	phosphate regulating endopeptidase homolog, X-linked	1	−1.0845	0.1616
201229	*LYRM9*	LYR motif containing 9	1	−1.8318	0.1709
6095	*RORA*	RAR-related orphan receptor A	1	−1.6688	0.1809
85439	*STON2*	stonin 2	2	−1.0335	0.2107
114781	*BTBD9*	BTB (POZ) domain containing 9	1	−1.4022	0.2145
144348	*ZNF664*	zinc finger protein 664	1	−1.1221	0.2262
27125	*AFF4*	AF4/FMR2 family, member 4	1	−1.3648	0.2620
152007	*GLIPR2*	GLI pathogenesis-related 2	1	−1.8377	0.2882
688	*KLF5*	Kruppel-like factor 5 (intestinal)	1	−1.0461	0.3321
27250	*PDCD4*	programmed cell death 4 (neoplastic transformation inhibitor)	1	−1.3298	0.3380
440295	*GOLGA6L9*	golgin A6 family-like 9	2	−1.5553	0.3385
55175	*KLHL11*	kelch-like family member 11	1	−1.1230	0.3808
85015	*USP45*	ubiquitin specific peptidase 45	1	−1.0026	0.3813
27109	*ATP5S*	ATP synthase, H+ transporting, mitochondrial Fo complex, subunit s (factor B)	1	−1.0441	0.3939
10802	*SEC24A*	SEC24 family member A	1	−1.1434	0.4081
2639	*GCDH*	glutaryl-CoA dehydrogenase	1	−1.2904	0.4110
843	*CASP10*	caspase 10, apoptosis-related cysteine peptidase	1	−1.1677	0.4282
8774	*NAPG*	N-ethylmaleimide-sensitive factor attachment protein, gamma	1	−1.1850	0.4452
54462	*CCSER2*	coiled-coil serine-rich protein 2	1	−1.3840	0.4566
9848	*MFAP3L*	microfibrillar-associated protein 3-like	1	−1.1114	0.4710
64764	*CREB3L2*	cAMP responsive element binding protein 3-like 2	1	−2.2962	0.4754
334	*APLP2*	amyloid beta (A4) precursor-like protein 2	1	−1.3719	0.5068
5163	*PDK1*	pyruvate dehydrogenase kinase, isozyme 1	1	−1.1852	0.5112
10124	*ARL4A*	ADP-ribosylation factor-like 4A	1	−1.4088	0.5161
145781	*GCOM1*	GRINL1A complex locus 1	1	−1.0388	0.5289
3987	*LIMS1*	LIM and senescent cell antigen-like domains 1	1	−1.0346	0.5454
57498	*KIDINS220*	kinase D-interacting substrate, 220kDa	1	−1.4304	0.5664
285636	*C5orf51*	chromosome 5 open reading frame 51	1	−1.0631	0.5666
9761	*MLEC*	malectin	1	−1.6201	0.5725
54477	*PLEKHA5*	pleckstrin homology domain containing, family A member 5	1	−1.3145	0.5846
10221	*TRIB1*	tribbles pseudokinase 1	1	−1.7224	0.5958
54014	*BRWD1*	bromodomain and WD repeat domain containing 1	1	−1.0694	0.6033
390	*RND3*	Rho family GTPase 3	1	−1.0681	0.6139
55823	*VPS11*	vacuolar protein sorting 11 homolog (S. cerevisiae)	1	−1.0569	0.6269
8444	*DYRK3*	dual-specificity tyrosine-(Y)-phosphorylation regulated kinase 3	1	−1.9227	0.6459
1978	*EIF4EBP1*	eukaryotic translation initiation factor 4E binding protein 1	1	−1.2832	0.6529
8874	*ARHGEF7*	Rho guanine nucleotide exchange factor (GEF) 7	1	−1.2069	0.6717
309	*ANXA6*	annexin A6	1	−1.9168	0.6821
5784	*PTPN14*	protein tyrosine phosphatase, non-receptor type 14	2	−1.2144	0.6888
100534599	*ISY1-RAB43*	ISY1-RAB43 readthrough	1	−2.4556	0.6928
54431	*DNAJC10*	DnaJ (Hsp40) homolog, subfamily C, member 10	2	−1.6506	0.7051
63874	*ABHD4*	abhydrolase domain containing 4	1	−1.6850	0.7071
196	*AHR*	aryl hydrocarbon receptor	1	−1.1288	0.7219
63897	*HEATR6*	HEAT repeat containing 6	1	−1.0711	0.7291
10961	*ERP29*	endoplasmic reticulum protein 29	1	−1.0578	0.7355
126626	*GABPB2*	GA binding protein transcription factor, beta subunit 2	2	−1.1151	0.7488
79794	*C12orf49*	chromosome 12 open reading frame 49	1	−1.7522	0.7864
5965	*RECQL*	RecQ helicase-like	3	−1.2203	0.7885
64651	*CSRNP1*	cysteine-serine-rich nuclear protein 1	1	−2.1079	0.8036
81558	*FAM117A*	family with sequence similarity 117, member A	1	−2.0458	0.8054
7706	*TRIM25*	tripartite motif containing 25	2	−1.2522	0.8360
55339	*WDR33*	WD repeat domain 33	1	−1.5940	0.8369
10097	*ACTR2*	ARP2 actin-related protein 2 homolog (yeast)	1	−1.0537	0.8586
23348	*DOCK9*	dedicator of cytokinesis 9	1	−1.2887	0.8621
10079	*ATP9A*	ATPase, class II, type 9A	1	−2.2117	0.8625
9497	*SLC4A7*	solute carrier family 4, sodium bicarbonate cotransporter, member 7	1	−1.1702	0.9121
54832	*VPS13C*	vacuolar protein sorting 13 homolog C (S. cerevisiae)	2	−1.1051	0.9176
23433	*RHOQ*	ras homolog family member Q	1	−1.6436	0.9319
55727	*BTBD7*	BTB (POZ) domain containing 7	1	−1.2974	0.9480
11260	*XPOT*	exportin, tRNA	1	−1.4398	0.9544
1362	*CPD*	carboxypeptidase D	2	−1.3036	0.9645
151887	*CCDC80*	coiled-coil domain containing 80	2	−2.1921	0.9667
116496	*FAM129A*	family with sequence similarity 129, member A	1	−1.8099	0.9903
9709	*HERPUD1*	homocysteine-inducible, endoplasmic reticulum stress-inducible, ubiquitin-like domain member 1	1	−3.3737	0.0854*
284723	*SLC25A34*	solute carrier family 25, member 34	1	−1.5051	0.0757*
83641	*FAM107B*	family with sequence similarity 107, member B	1	−2.2173	0.0749*
60412	*EXOC4*	exocyst complex component 4	1	−1.0957	0.0691*
6700	*SPRR2A*	small proline-rich protein 2A	1	−2.1995	0.0568*
10365	*KLF2*	Kruppel-like factor 2	1	−1.3378	0.0386*
3572	*IL6ST*	interleukin 6 signal transducer	1	−1.3786	0.0330*
10551	*AGR2*	anterior gradient 2	1	−3.7365	0.0134*
9663	*LPIN2*	lipin 2	1	−2.6332	0.0033*
155435	*RBM33*	RNA binding motif protein 33	1	−1.3082	0.0029*
54855	*FAM46C*	family with sequence similarity 46, member C	2	−1.0118	0.0028*
728661	*SLC35E2B*	solute carrier family 35, member E2B	1	−1.0811	0.0004*
23591	*FAM215A*	family with sequence similarity 215, member A (non-protein coding)	1	−1.4327	N/A
643707	*GOLGA6L4*	golgin A6 family-like 4	2	−1.3773	N/A

* poor prognosis in patients with low gene expression.

**Table 3 cells-08-01535-t003:** Clinical features of 9 HNSCC cases used for immunohistochemical staining.

	Age	Sex	Location	T	N	M	Stage	Differentiation
A	80	M	larynx	3	2c	0	IVa	moderate
B	73	M	larynx	3	0	0	III	poor
C	77	M	oral	2	2b	0	Iva	moderate
D	42	F	oral	4a	0	0	IVa	poor
E	51	M	oral	2	0	0	II	well
F	52	F	oral	4a	2c	1	Ivc	well
G	72	M	hypopharynx	2	0	0	II	moderate
H	64	M	hypopharynx	2	2b	0	IVa	well
I	70	M	hypopharynx	2	2b	0	Iva	well

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
