# Peer review of "Regulation of Oncogenic Targets by miR-99a-3p (Passenger Strand of miR-99a-Duplex) in Head and Neck Squamous Cell Carcinoma"

_cells, 2019, doi:10.3390/cells8121535_

Round 1

Reviewer 1 Report

This is a study of miRNAs in squamous cell carcinoma of the head and neck.

MAJOR COMMENTS

This is a heterogeneous group of tumours, and previous results have shown a differnce in expression of proteins and miRNAs based on sub site. The fact that this study mixes hypopharyngeal, laryngeal, tongue and floor of mouth tumours is thus not correct. It does give a bigger group of tumours analysed but in the analyses tumours must be separated based on sub site, which thus gives too small groups for any valid conclusions.

Reviewer 2 Report

Brief summary:

In this manuscript authors describe the oncogenic role of the passenger strand miR-99a-3p in head and neck squamous cell carcinoma (HNSCC). On the one hand, they demonstrated that, far from the null functionality, passenger strands of miRNAs can play a relevant role in carcinogenesis. They also demonstrated that in the case of the miR-99a-duplex both strands (miR-99a-5p and miR-99-3p) acted as antitumor miRNAs in HNSCC cells. They have identified oncogenic targets regulated by miR-99a-3p, demonstrating the prognostic value of some of them. At last, the authors have studied the role of STAMBP (a miR-99a-3p regulated gene) in proliferation, migration and invasion of HBSCC cells, suggesting that it could promote cancer cell metastasis.

Broad comments:

After careful reading of the manuscript, this reviewer can say that the authors have carried out a well-planned and organized work. It is worth recognizing the laborious work of getting samples from patients, although I believe that they should get more results from the above mentioned samples.

1.- Results 2.1.: when comparing expression levels of miR-99a-5p and miR-99-3p in cancer tissues with those in normal tissues, which type of normal tissue have you analyzed? I assume that you compare cancerous and normal tissue from the same individuals. It should be explained in the text.

2.- Results 2.1. Figure 1D: In the cohort from the Cancer Genome Atlas, how do you set the threshold for low and high miRNA expression? In this cohort, did you analyze the correlation between miR-99a-5p and miR-99-3p? Is there a positive correlation as shown in Figure 1C?

3.- Results 2.2. It would be interesting to see the results shown in Figure 2 and Figure 5 when transferring the two miRNAs (miR-99a-5p and miR-99-3p) at the same time. Have you tried it? Why?

4.- Results 2.5 and 2.6. When you studied the clinical significance of miR-99a-3p targets in HNSCC pathogenesis you found that, among the list of 114 genes, high expression of 10 of them was associated with poor prognosis. Furthermore, you elucidated that expression levels of 5 of them were independent prognostic factors for 5-year overall survival. However, you have only deepened the study in STAMBP. What do you know about the regulation of the other genes by miR-99a-3p? In my opinion at least the mRNA and protein levels of TIMP4, TMEM14C, SUV420H1 and CANX in miR-99a-5p and miR-99-3p transfected FaDu and SAS cells should be included in the work.

5.- Results 2.6. Figure 5. What about the results of the luciferase assay in FaDu cells?

6.- Results 2.7. In my opinion, the proliferation results obtained with FaDu cells compared to SAS cells have to be discussed.

7.- Results 2.8. Overexpression of STAMBP in HNSCC clinical specimens is shown. It is necessary to show the STAMBP expression in healthy tissues. Moreover, it would be interesting to quantify the expression of STAMBP in each of the 22 samples and to show that result comparing it with the expression observed in the 22 healthy tissues.

8.- Material and Methods. Although references of works describing the methodology are shown, a brief description of the functional assays performed, the immunoprecipitation assay, sequences of transfected vectors, etc. should be included in the manuscript. Information about healthy tissue samples is also missing.

Specific comments:

- Introduction. In the fourth paragraph (line 66 to 73) it should be added that although the traditional concept of passenger strand refers to non-functional miRNAs, it is currently known that some passenger miRNA strands exert antitumor roles. I believe that what is commented in the discussion (lines 219-222) would also have to be included in the introduction to justify studying the role of the miR-99a-3p in HNSCC pathogenesis. – Results, Figure 1. The figure legend is not correct. Please check that the letters in the figure legend (A,B,C,D) coincide with the graphs in the figure . – Results, Table 1. Authors should add a legend explaining the meaning of some abbreviations included in the top of the table (T, N, M). – Figure 3S. I would say that the gene selection process for miR-99a-3p shown in Figure 3S has a mistake. In the second box, instead of “Gere expression analysis in miR-99a-5p transfected FaDu cells (GSE123318) (log2 ratio < -1.0) 1286 genes”, it should be “Gene expression analysis in miR-99a-3p transfected FaDu cells (GSE123318) (log2 ratio < -1.0) 1286 genes”.

Round 2

Reviewer 2 Report

Brief summary:

In this manuscript authors describe the oncogenic role of the passenger strand miR-99a-3p in head and neck squamous cell carcinoma (HNSCC). On the one hand, they demonstrated that, far from the null functionality, passenger strands of miRNAs can play a relevant role in carcinogenesis. They also demonstrated that in the case of the miR-99a-duplex both strands (miR-99a-5p and miR-99-3p) acted as antitumor miRNAs in HNSCC cells. They have identified oncogenic targets regulated by miR-99a-3p, demonstrating the prognostic value of some of them. At last, the authors have studied the role of STAMBP (a miR-99a-3p regulated gene) in proliferation, migration and invasion of HBSCC cells, suggesting that it could promote cancer cell metastasis.

Broad comments:

After careful reading of the manuscript, this reviewer can say that the work has improved with the changes made. In any case, this reviewer considers that there are things that can be improved, especially regarding the format.

1.- Introduction: I considered that the text should be reorganized between lines 65-78. The text that has been introduced in the new version would be better within the following paragraph, and not as a separate paragraph

2.- Figure 1: Now thefigure legend is correct, however, the references to the different sections of figure 1 in the text are not correct.

3.- Results from the Figure S8 should be discussed much more.
